# Apoptotic Cell Death in Bicuspid-Aortic-Valve-Associated Aortopathy

**DOI:** 10.3390/ijms24087429

**Published:** 2023-04-18

**Authors:** Sarah J. Barnard, Josephina Haunschild, Linda Heiser, Maja T. Dieterlen, Kristin Klaeske, Michael A. Borger, Christian D. Etz

**Affiliations:** 1Heisenberg Working Group, Saxonian Incubator for Clinical Translation, Philipp-Rosenthal-Str. 55, 04103 Leipzig, Germany; 2University Department for Cardiac Surgery, Heart Center Leipzig, 04289 Leipzig, Germany

**Keywords:** vascular biology, aneurysm, vascular disease, cell signaling, bicuspid aortic valve

## Abstract

The bicuspid aortic valve (BAV) is the most common cardiovascular congenital abnormality and is frequently associated with proximal aortopathy. We analyzed the tissues of patients with bicuspid and tricuspid aortic valve (TAV) regarding the protein expression of the receptor for advanced glycation products (RAGE) and its ligands, the advanced glycation end products (AGE), as well as the S100 calcium-binding protein A6 (S100A6). Since S100A6 overexpression attenuates cardiomyocyte apoptosis, we investigated the diverse pathways of apoptosis and autophagic cell death in the human ascending aortic specimen of 57 and 49 patients with BAV and TAV morphology, respectively, to identify differences and explanations for the higher risk of patients with BAV for severe cardiovascular diseases. We found significantly increased levels of RAGE, AGE and S100A6 in the aortic tissue of bicuspid patients which may promote apoptosis via the upregulation of caspase-3 activity. Although increased caspase-3 activity was not detected in BAV patients, increased protein expression of the 48 kDa fragment of vimentin was detected. mTOR as a downstream protein of Akt was significantly higher in patients with BAV, whereas Bcl-2 was increased in patients with TAV, assuming a better protection against apoptosis. The autophagy-related proteins p62 and ERK1/2 were increased in patients with BAV, assuming that cells in bicuspid tissue are more likely to undergo apoptotic cell death leading to changes in the wall and finally to aortopathies. We provide first-hand evidence of increased apoptotic cell death in the aortic tissue of BAV patients which may thus provide an explanation for the increased risk of structural aortic wall deficiency possibly underlying aortic aneurysm formation or acute dissection.

## 1. Introduction

The bicuspid aortic valve (BAV) is the most common congenital cardiovascular malformation affecting 1–2% of the general population. Characteristically, two of the three leaflets of the aortic valve are fused, resulting in a valve with only two functioning leaflets [1]. Compared to patients with a tricuspid aortic valve (TAV), patients with BAV are more frequently male and have a higher risk of associated pathologies such as proximal aortic dilatation, aneurysm formation at an early age and acute dissection [2]. It is widely accepted that the composition of the extracellular matrix is different in ascending aortic aneurysms of BAV compared to TAV patients [3,4]. In BAV patients, the matrix metalloproteinase 2 (MMP-2) RNA and protein expression are increased [5]. Fukami et al. showed that rat-sarcoma-mediated MMP-2 activation is induced by advanced glycation end (AGE) products and their receptor (RAGE) [6]. RAGE is highly expressed during brain development, but also present in cell types such as monocytes/macrophages, endothelial cells, mesangial cells and smooth muscle cells [7,8]. Moreover, Branchetti and coworkers investigated the levels of circulating soluble RAGE (sRAGE) in the plasma of BAV and TAV patients, and detected a direct correlation between sRAGE level and the appearance of BAV and BAV-associated aortopathies [9]. Another interacting partner of RAGE is the S100 calcium-binding protein A6 (S100A6), which is expressed in several tissues such as smooth muscle, brain, lung, kidney and heart [10,11]. It regulates different intracellular functions such as cell proliferation, differentiation, migration and apoptosis [12].

### 1.1. Apoptosis

Apoptosis is a form of programmed cell death mediating development and aging, but it is also a homeostatic mechanism that can maintain cell populations in diverse tissues, act as a defense mechanism in immune reactions and remove damaged cells in disease [13]. It is a balanced mechanism to remove components or damaged cells from an organism without harming or destroying the organism and is characterized by morphological and biochemical changes in the cell [14,15]. The initiation of apoptosis is mediated by the activation of caspases, which are cysteine-aspartic proteases. Three mechanisms are known to play a role in apoptosis: an extrinsic receptor–ligand-mediated mechanism that activates caspase-8, an intrinsic mitochondrial or B-cell lymphoma 2 (Bcl-2)-regulated pathway that activates caspase-9 and a cascade involving the endoplasmic reticulum that activates caspase-12 [16,17]. Both the receptor–ligand-mediated mechanism and the mitochondrial pathway may lead to the activation of caspase-3 [18]. 

Byun et al. showed that caspase-3 and -7 cleave the intermediate filament protein vimentin in HeLa cells. The 57 kDa vimentin is cleaved in aspartic acid at position 85 (Asp^85^), generating a 48 kDa fragment resulting in the dismantling of the intermediate filaments into punctate or granular aggregates [19]. A further substrate for caspase-3 is α-smooth muscle actin. In myofibroblasts, caspase-3 cleaves α-smooth muscle actin and thus promotes the reorganization of the cytoskeleton, leading to apoptosis [20].

While caspases are pro-apoptotic proteins, others such as Bcl-2 are anti-apoptotic and therefore pro-survival. Through direct binding to pro-apoptotic proteins such as the Bcl-2-antagonist of cell death or the Bcl-2-associated X protein, Bcl-2 inhibits cell death [21]. Another cell survival regulator in several cell types during apoptosis is Akt. It is activated by phosphoinositide 3-kinase (PI3K) and blocks apoptosis probably via phosphorylation and the inhibition of pro-apoptotic proteins [22]. Activated Akt decreases the tuberous sclerosis complex subunit 2 (TSC2) expression followed by the activation of the mechanistic target of rapamycin (mTOR) toward p70 ribosomal protein S6 kinase (p70S6K) and eukaryotic translation initiation factor 4E-binding protein 1 (4E-BP1). Both p70S6K and 4E-BP1 regulate protein translation and apoptosis [23,24].

### 1.2. Autophagic Cell Death

In contrast to apoptosis resulting in cell death, autophagic cell death can contribute to cell death or cell survival which is fully dependent on the stimuli and cellular context [25]. It functions to recycle damaged or misfolded proteins, organelles and aggregates. The autophagy-related protein (Atg) 5 complex, class III phosphatidylinositol 3-kinase (PI3K)/vacuolar protein sorting 34 complex and microtubule-associated protein 1 light chain 3 (LC3, the main mammalian homolog of Atg8)-II contribute to form the mature double-membrane autophagosome. Via the fusion of an autophagosome with a lysosome, an autophagolysosome is generated. Within the autophagolysosome, the elements to be recycled are degraded and removed by lysosomal hydrolases. Valuable building blocks of amino acids, fatty acids and simple sugars are released into the cytosol and can be used for protein synthesis and adenosine triphosphate production. Thus, autophagic cell death regulates the balance between the production and recycling of cellular components [26,27].

Another prominent marker of autophagy activity is nucleoporin 62 (p62). It directly binds to LC3 and is a substrate during autophagic degradation [28,29]. Amongst others, p62 transcription is regulated by the rat sarcoma/mitogen-activated protein kinase (MAPK) pathway. One key protein of the cascade is the extracellular signal-regulated kinases (ERK) which can hetero-oligomerize with p62 and colocalize with LC3-II [30,31]. 

In this study, we analyzed the tissues of patients with BAV and TAV regarding the protein expression of RAGE and its ligands AGE and S100A6. Since it is known that S100A6 overexpression attenuates cardiomyocyte apoptosis [32], we furthermore investigated the proteins of the diverse pathways of intrinsic apoptosis and autophagy to identify differences and explanations for the higher risk of patients with BAV for severe diseases such as proximal aortic dilatation, aneurysm or acute type A dissection.

## 2. Results

### 2.1. Expression of AGE, RAGE and S100A6

Based on the published correlation between sRAGE level and the appearance of BAV and BAV-associated aortopathies [9], we analyzed the protein expression levels of RAGE and its ligands AGE and S100A6 via Western blot analyses and ELISA, respectively, in the whole-tissue lysates of patients with BAV and TAV (Figure 1) morphology, irrespective of their functional valve status (aortic stenosis/aortic regurgitation). 

The relative protein expression of RAGE was significantly increased in the tissue of patients with BAV compared to TAV (Figure 1). Within the convex regions of the aorta, no differences between patients with BAV and TAV were detected, whereas the concave site of the aorta of patients with BAV showed a significantly higher RAGE protein expression than the concave site of the aorta of patients with TAV (Figure 1A; Table 1). The protein expressions of the ligands AGE and S100A6 were increased in the tissues of patients with BAV in both aortic regions (Figure 1B,C; Table 1). The results of the AGE and RAGE protein expression were verified using ELISA. 

### 2.2. Protein Expression of Pro-Apoptotic Proteins 

To investigate the mechanism of apoptosis, we performed Western blot analysis of the pro-apoptotic proteins caspase-3, vimentin and α-smooth muscle actin (Figure 2; Table 2). In Figure 2A, the ratio of the active subunits of caspase-3 (17 kDa and 12 kDa), which leads to the degradation of several cellular proteins, morphological changes and the induction of apoptosis and mature inactive caspase-3 (32 kDa), is shown. The ratio was significantly higher in the tissues of patients with TAV than in patients with BAV, suggesting higher apoptotic activity of caspase-3 in patients with TAV. Furthermore, there were no differences in the convex sites, but there was a significant increase detected in the ratio of the active and mature caspase-3 in the TAV concave region compared to the BAV concave region. 

Next, we performed Western blot analyses of the full-length intermediate filament protein vimentin (57 kDa), which is cleaved by caspase-3, generating a 48 kDa fragment. Regarding the relative protein expression of the full-length vimentin, no differences in the tissues of patients with BAV and TAV were observed. In contrast, the relative protein expression of the 48 kDa vimentin fragment was slightly increased in the tissues of the BAV patients. An increase in cleaved vimentin protein expression in the convex region of the samples of the BAV patients compared to the TAV convex regions was detected. The concave regions displayed no differences (Figure 2B).

The relative protein expression of α-smooth muscle actin, a further substrate of caspase-3, was investigated. Even though less active caspase-3 was detected in the tissue of patients with BAV, the relative protein expression of α-smooth muscle actin in the same samples increased 1.5-fold compared to the tissue of TAV patients. Separated in the convex and concave regions of the aorta, patients with BAV showed an increase in α-smooth muscle actin protein expression in both regions (Figure 2C). 

### 2.3. Protein Expression Levels of Anti-Apoptotic Proteins 

We further investigated the relative protein expression of the anti-apoptotic proteins Bcl-2, Akt and mTOR via Western blot analyses (Figure 3; Table 3). For Bcl-2, we detected a significant increase in protein expression in the tissue of patients with TAV, especially in the concave region of the aorta. In the convex region, no alterations were found (Figure 3A). 

For the cell survival regulator Akt, no appreciable difference was found in the relative protein expression between patients with BAV or TAV, nor between the different regions of the aorta (Figure 3B). For mTOR, a downstream protein of Akt leading to apoptosis and significantly elevated protein expression in the tissues of patients with BAV were detected. This increase was present in the concave but not in the convex region of the aorta (Figure 3C).

### 2.4. Investigation of Autophagy-Linked Proteins LC3 and p62 

As many articles have published that there is a crosstalk between apoptosis and autophagic cell death, we also studied proteins in pathways leading to autophagic cell death (Figure 4; Table 4). 

The ratio of the relative protein expression of active LC3-II and mature LC3-I revealed a minor increase and thus a higher amount of autophagosomes in the tissue of patients with TAV in the convex region of the aorta (Figure 4A). The protein expression of p62 was increased in the tissues of BAV patients. An investigation int the different aortic regions showed a significant increase in protein expression on the concave site of patients with BAV, whereas no differences on the convex site could be observed (Figure 4B). Similar observations could be made for ERK1/2. The protein expression was significantly elevated in the tissues of patients with BAV. The different regions also showed a non-significant increase on the convex and a more pronounced elevation at the concave site of the aorta (Figure 4C).

## 3. Discussion

In the present study, we analyzed ascending aortic tissues of patients with aortic aneurysm and BAV or TAV morphology with regard to selected proteins of signal transduction pathways leading to apoptosis or autophagic cell death. Apoptosis is induced by increased caspase-3 activity, which leads to elevated levels of the 48 kDa fragment of vimentin, as well as α-smooth muscle actin [18,19,20]. This is possibly associated with RAGE and its ligands resulting in weakening the aortic wall and making it prone to the development of proximal aortic aneurysms, acute aortic dissection or rupture. Anti-apoptotic proteins such as Bcl-2, Akt and mTOR can protect cells from apoptosis [21,22,23,24]. Autophagic cell death can be defined by the ratio of LC3II to LC3I, as well as its interaction partner p62. ERK, another interaction partner of p62, can activate autophagic cell death by inducing LC3, as well as upregulate pro-apoptotic proteins and downregulate anti-apoptotic proteins [26,27,28,29,30].

### 3.1. Increased Expression of RAGE and Its Ligands AGE and S100A6

In 2014, Branchetti et al. described that circulating sRAGE is closely related to BAV and its associated aortopathy [9]. It is known that the aortic wall is infiltrated by circulating glycation products via AGE-mediated endothelial hyperpermeability. This leads to the synthetic phenotype of vascular smooth muscle cells (VSMCs) and the remodeling of the extracellular matrix in vivo and ex vivo [33]. Our study demonstrates a significant increase in the relative protein expression levels of both total RAGE and its ligands AGE and S100A6, in patients with BAV. As EF-hand calcium-binding proteins, S100 proteins regulate different intracellular functions such as cell proliferation, differentiation, migration and apoptosis. The interaction of RAGE and S100 induces cellular signaling [12]. It is known that S100A6 promotes apoptosis in Hep3B cells via the upregulation of caspase-3 activity [34]. In Hep-2 cells, S100A6 expression leads to higher tumor protein P53 transcriptional activity, also resulting in apoptosis [35]. In contrast, in rat cardiac myocytes, the expression of S100A6 limited apoptosis and infarct size after myocardial ischemia-reperfusion in an experimental rat model [32,36]. We demonstrated that the protein expression levels of RAGE and its ligands AGE and S100A6 are elevated in the aortic tissue of patients with BAV, which may promote apoptosis via the upregulation of caspase-3 activity. To see whether the increase in S100A6 promotes or attenuates apoptosis, we further analyzed vimentin and α-smooth muscle actin.

### 3.2. Protein Expression of Pro-Apoptotic Proteins Suggesting a Higher Affinity for Apoptosis in the Tissues of Patients with BAV

Caspases are the most important players in the apoptosis machinery and are functionally subdivided into initiator (caspase-8, -9 and -10) and effector (caspase-3, -6 and -7) caspases [37]. All caspases are expressed as proenzymes 30–50 kDa in size, containing two subunits of approximately 20 kDa and 10 kDa, respectively. For their activation, proteolytic processing between domains is necessary, and finally, both subunits need to form a heterodimer [38]. In turn, activated caspases proteolytically cleave a wide variety of cellular proteins of diverse signal cascades, resulting in cell death [39]. The two crucial caspases during apoptosis are caspase-3 and caspase-7. Caspase-3 is required for DNA fragmentation and morphological changes and caspase-7 is responsible for apoptotic cell detachment from the extracellular matrix [40,41].

In 2010, it was published that the inhibition of caspase-3 protected in vitro cultured VSMC of the ascending aorta against calcium chloride apoptosis significantly more in the concave site than in the convex site [42]. However, in our study, significantly increased caspase-3 activity was detected in tissues from patients with TAV, mainly in the concave site. One reason for this discrepancy could be the preparation of the samples. While Mohamed et al. worked with isolated VSMCs, whole cell lysates, including aortic endothelial cells, aortic smooth muscle cells and aortic adventitial fibroblasts, were used in our experiments. The different cell types could have different caspase-3 activities. It is also known that caspase-3 activation is dependent on stimulus and cell type [43,44].

Another important protein in the induction of apoptosis is vimentin, whose cleavage to a 48 kDa fragment by caspase-3 leads to the dismantling of the intermediate filaments. These morphological changes occur at the same time that the nuclear fragmentation in apoptosis is initiated, demonstrating that the vimentin proteolysis by caspase-3 and -7 is sufficient to induce apoptosis. The cleaved proteolytic fragment activates more caspases and thereby amplifies the cell death signal by generating a positive feedback loop [19]. We also showed that the relative protein expression of the 48 kDa vimentin fragment is increased in patients with BAV compared to patients with TAV. This increase was seen in both the concave and the convex site of the aorta, potentially activated by changes in blood architecture, mainly vortex formation, as seen via 4D flow MR imaging [45]. It is also proposed that vimentin as a mechanosensitive regulator of Notch signaling leads to the structural remodeling of the arterial wall in response to changes in hemodynamic conditions [46]. This leads to the assumption that patients with BAV are prone to apoptosis. This hypothesis is supported by the results of the relative protein expression of α-smooth muscle actin, which showed a significant increase in patients with BAV. Comparing the different sites of the aorta, we could demonstrate an increase in the relative protein expression of α-smooth muscle actin on both sites of the aorta. Interestingly, the protein expression is significantly increased in the concave site. In myofibroblasts, the contractile activity is terminated in remodeling such as during wound healing when the tissue is repaired. In this context, α-smooth muscle actin expression decreases and myofibroblasts are degraded by apoptosis [47]. Applied to our results, this would mean that the lower α-smooth muscle actin expression in patients with TAV results in apoptosis. Furthermore, the integration of α-smooth muscle actin into stress fibers significantly enhances the contractile activity of fibroblastic cells [48]. The significant increase in α-smooth muscle actin expression on the concave site of the aorta of patients with BAV could thus be associated with a strengthening of the extracellular matrix and explain why aneurysms mostly develop on the convex site [49].

### 3.3. Protein Expression Levels of Anti-Apoptotic Proteins Imply a Protection from Apoptosis in the Tissue of Patients with TAV

By analyzing the relative protein expression of anti-apoptotic proteins, we showed that patients with TAV had significantly more Bcl-2. However, whilst no difference was found on the convex site, there was a significant elevation on the concave site of patients with TAV. Assuming that Bcl-2 decreases apoptosis mainly by preventing mitochondrial outer membrane permeabilization via neutralizing the activity of pro-apoptotic members such as the Bcl-2-antagonist of cell death or the Bcl-2-associated X protein, it is likely that patients with TAV are better protected against apoptosis [50]. 

It has already been described that Akt is a major regulator in survival signaling. It is phosphorylated by PI3K and thus activated and disassociated from the membrane. Activated Akt phosphorylates/activates other pro-survival proteins or inactivates pro-apoptotic signaling cascades [51]. In our experiments, we did not detect any significant differences in the relative protein expression of Akt in patients with BAV or TAV. It is conceivable that the regulation of survival signaling is only visible via Akt phosphorylation and not by total Akt protein expression.

In the absence of Akt activity, the tuberous sclerosis complex subunits 1 and 2 (TSC1 and 2) suppress the activation of mTOR. Once activated, Akt decreases the TSC2 expression, thus activating mTOR [23,24]. The activities of mTOR complex 1 (mTORC1) and mTORC2 were estimated using the phosphorylation states of ribosomal S6 protein and Akt, respectively. The inhibition of mTOR via rapamycin revealed reduced apoptosis and promoted autophagy in cardiomyocytes by regulating the crosstalk between the mTOR and endoplasmic reticulum (ER) stress pathways in chronic heart failure [52]. Due to the significantly increased relative mTOR protein expression in patients with BAV, it can be assumed that there is elevated apoptosis activity, but possibly decreased autophagy activity. It was shown that shear stress can regulate mTOR and its activation is necessary to activate p70S6K and thus cell growth. The PI3K/Akt/mTOR/p70S6k pathway may play an important role in shear-stress-induced migration, phenotypic transformation, apoptosis and the proliferation of VSMCs [53]. The comparison of the convex sites of the aortas showed no differences, but we found a significant increase in the relative protein expression of mTOR on the concave site of aortas from patients with BAV. This finding may also support the hypothesis that the concave site of the aorta of patients with BAV is better protected from autophagy than the convex site and that aneurysm formation may occur at the convex site. 

### 3.4. Autophagic Cell Death Is More Likely to Occur in Patients with BAV

A good indicator for monitoring autophagy is LC3. Two post-translationally produced isoforms of LC3 are known: LC3-I and LC3-II. While LC3-I is localized in the cytoplasm, LC3-II is associated with membrane compartments involved in the autophagic pathway. The amount of LC3-II corresponds to the number of autophagosomes [54]. With regard to our results, this would mean a minimally increased number of autophagosomes in the tissue of patients with TAV and especially on the convex site of the aorta. Only based on the LC3II/LC3I ratio, no statement can be made regarding an increased autophagic cell death risk for patients with neither BAV nor TAV.

A direct interacting partner of LC3 is p62. Via degradation, autophagy decreases the intracellular level of p62. In contrast, p62 can activate mTORC1, thereby suppressing autophagy followed by an accumulation of p62 levels [28,29]. This hypothesis is consistent with our findings that elevated p62 levels in patients with BAV, particularly on the concave site, can cause an increase in relative mTOR protein expression in the same tissue segments. As p62 is an autophagy substrate and delivers ubiquitinated cargoes for autophagic degradation, it can be used as a reporter of autophagy activity [30,55]. With our experiments, we were able to show a considerable increase in the relative p62 protein expression in patients with BAV. While the relative protein expression is identical regarding the convexity, there is a significant increase at the concave site in patients with BAV compared to TAV, leading to the assumption of higher autophagic cell death activity. On the other hand, Paine and coworkers demonstrated that an overexpression of p62 in HEK cells results in aggregate formation that may protect cells from apoptosis [56]. A critical protein of the MAPK pathway that regulates the p62 transcription is ERK. It hetero-oligomerizes with p62 [30]. Our experiments showed a similar relative protein expression of ERK1/2 compared with that of p62. In different cell lines (L929, MCF-9 or WI38), ERK-dependent autophagic activity is also associated with the induction of LC3 and the conversion of LC3-I to LC3-II [57,58,59]. From our results, it can be hypothesized that patients with BAV show increased autophagic cell death activity. It has also been published that ERK is associated with the intrinsic apoptotic pathway characterized by the release of cytochrome c from mitochondria and the activation of initiator caspase-9, or with the extrinsic apoptotic pathway, which is based on the activation of caspase-8 and following caspase-3. Activated ERK is also able to upregulate pro-apoptotic proteins, such as Bax, and downregulate anti-apoptotic proteins, such as Bcl-2 [60]. This is reflected in our Bcl-2 collected data.

## 4. Materials and Methods

### 4.1. Patient Population and Sample Collection

The study was approved by the local ethics committee of Leipzig University, Faculty of Medicine (approval no. 177/15-ek), on 3 June 2015. Patients were included after they gave written informed consent. Patients with connective tissue disorder, acute infection (e.g., endocarditis) or acute aortic syndrome were excluded. Patients’ characteristics are listed in Table 5.

Samples from the ascending aorta of patients were collected during ascending aortic replacement, and aortic valve morphology was classified as TAV or BAV. Aortic specimens were dissected directly after removal and subdivided into convexity and concavity (Figure 5), snap frozen in liquid nitrogen and stored at −80 °C until biomolecular analyses were performed.

### 4.2. Preparation of Protein Lysates

For the preparation of protein lysates from human aorta, 30–50 mg tissue was homogenized in 400 µL RIPA Buffer (Thermo Fisher Scientific, Dreieich, Germany) using a TissueLyser II (Qiagen, Hilden, Germany). The homogenate was centrifuged for ten min at 12,000× *g* at 4 °C. The supernatant was transferred into a new reaction tube and the protein concentration was measured using the Pierce^TM^ BCA Protein Assay Kit (Thermo Fisher Scientific, Dreieich, Germany). 

### 4.3. Western Blot Analysis

All antibodies were obtained from standard suppliers and are attached in the Appendix A.

For Western blot analyses, 20 µg of lysates was separated in 12% SDS-PAGE. The proteins were transferred to a 0.45 µm polyvinylidene difluoride membrane (Roth, Karlsruhe, Germany) and incubated with primary antibodies overnight at 4 °C as well as horseradish peroxidase (HRP)-conjugated secondary antibodies for 1 h at room temperature (see Appendix A). Proteins were detected using the SERVALight Vega CL HRP WB Substrate Kit (Serva, Heidelberg, Germany) and Fusion-FX7 (Vilber Lourmat, Eberhardzell, Germany). GAPDH and α-Tubulin served as the loading controls.

### 4.4. Enzyme-Linked Immunosorbent Assay (ELISA)

For the quantification of AGE, RAGE and S100A6, the Quantikine^®^ ELISA Human RAGE Immunoassay (R&D Systems^®^, Minneapolis, MN, USA), Human AGEs ELISA Kit (Abbexa, Cambridge, UK) and S100A6 ELISA Kit (Cloud-Clone Corp., Katy, TX, USA), respectively, were used. All assays were performed according to the manufacturer’s instructions. The optical density was measured using a Synergy H1 Microplate Reader (BioTek, Berlin, Germany). 

### 4.5. Statistical Analyses

To semi-quantify the intensities of the Western blot analyses, ImageJ 1.49 (developed by Wayne Rasband) was used. The intensities of the proteins of interest were normalized to the loading controls. Statistical analyses were performed using Prism 6 (GraphPad Software, La Jolla, CA, USA). The Gaussian distribution was tested using the Kolmogorov–Smirnov and Shapiro–Wilk normality tests. Differences in protein expression or activities between the groups were analyzed using Student’s *t*-test. Statistical significance was indicated by a *p*-value < 0.05. 

## 5. Conclusions

In conclusion, based on the collected data from our experiments with tissues of 106 patients, many factors provide first-hand evidence of increased apoptotic cell death in tissues from patients with BAV and an explanation for the increased risk of diseases such as aortic dilatation, aneurysm or dissection (Appendix A). A significantly increased protein expression of RAGE, AGE and S100A6 in patients with BAV may promote apoptosis and lead to the remodeling of the extracellular matrix. Although increased caspase-3 activity was not detected in patients with BAV, an increased protein expression of the 48 kDa fragment of vimentin was detected. The increased protein expression of α-smooth muscle actin in the concave region of the bicuspid aorta could lead to a strengthening of the extracellular matrix and explain why aneurysms occur more frequently in the convex region. This is supported by the elevated mTOR protein expression in the concave region, leading to cell growth and therefore protection from apoptosis. 

In contrast, the evidence of autophagic cell death is not as strong. No differences in the LC3II/LC3I ratio could be detected between patients with BAV and TAV, respectively. The interacting partner p62 showed increased levels that could activate mTORC1 and thereby suppress autophagy. ERK, which can hetero-oligomerize with p62, leads to increased activity of pro-apoptotic proteins, assuming that cells in bicuspid tissue are more likely to undergo apoptotic cell death leading to changes in the cell wall and finally to severe aortopathies.

## Figures and Tables

**Figure 1 ijms-24-07429-f001:**
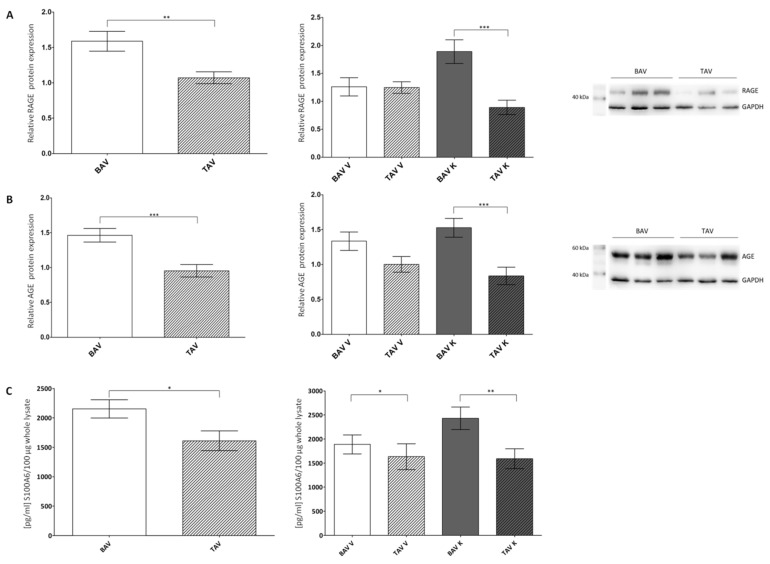
Statistical analysis of protein expression levels of RAGE and its ligands AGE and S100A6 in whole-tissue lysates of patients with BAV and TAV from the convex (V) and concave (K) regions of the aorta. (**A**) Relative RAGE protein expression via Western blot analysis in the tissue of patients with BAV (*n* = 69) and TAV (*n* = 59) and separated by regions BAV V (*n* = 33), BAV K (*n* = 36), TAV V (*n* = 30) and TAV K (*n* = 29). GAPDH served as the loading control. (**B**) Relative AGE protein expression via Western blot analysis in the tissue of patients with BAV (*n* = 69) and TAV (*n* = 59) and separated by regions BAV V (*n* = 33), BAV K (*n* = 36), TAV V (*n* = 30) and TAV K (*n* = 29). GAPDH served as the loading control. (**C**) Quantification of S100A6 protein expression using ELISA in the tissue of patients with BAV (*n* = 57) and TAV (*n* = 89) and separated by regions BAV V (*n* = 29), BAV K (*n* = 28), TAV V (*n* = 42) and TAV K (*n* = 47). *p* values of Student’s *t*-test: * *p* < 0.05, ** *p* < 0.01, *** *p* < 0.0001.

**Figure 2 ijms-24-07429-f002:**
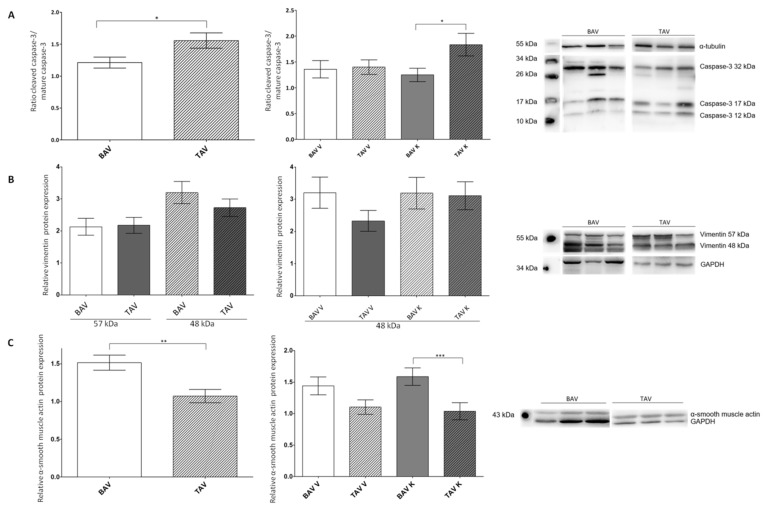
Statistical analysis of the protein expression levels of the pro-apoptotic proteins caspase-3, vimentin and α-smooth muscle actin in whole-tissue lysates of patients with BAV and TAV from the convex (V) and concave (K) regions of the aorta. (**A**) Ratio of protein expression of cleaved active caspase-3 and mature inactive caspase-3 via Western blot analysis in the tissue of patients with BAV (*n* = 97) and TAV (*n* = 87) and separated by regions BAV V (*n* = 50), BAV K (*n* = 47), TAV V (*n* = 42) and TAV K (*n* = 45). α-tubulin served as the loading control. (**B**) Relative vimentin protein expression (57 kDa/48 kDa) via Western blot analysis in the tissue of patients with BAV (*n* = 99/*n* = 96) and TAV (*n* = 94/*n* = 93) and 48 kDa vimentin protein expression separated by regions BAV V (*n* = 48), BAV K (*n* = 48), TAV V (*n* = 46) and TAV K (*n* = 47). GAPDH served as the loading control. (**C**) Relative α-smooth muscle actin protein expression via Western blot analysis in the tissue of patients with BAV (*n* = 104) and TAV (*n* = 86) and separated by regions BAV V (*n* = 52), BAV K (*n* = 52), TAV V (*n* = 44) and TAV K (*n* = 42). GAPDH served as the loading control. *p* values of Student’s *t*-test: * *p* < 0.05, ** *p* < 0.01, *** *p* < 0.0001.

**Figure 3 ijms-24-07429-f003:**
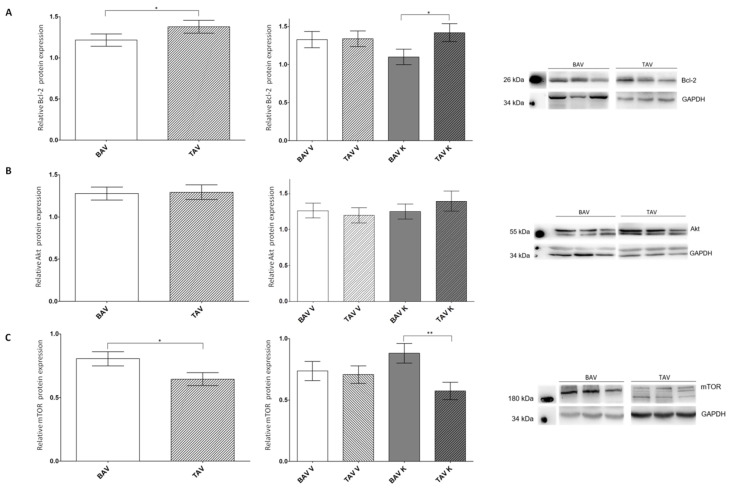
Statistical analysis of protein expression levels of the anti-apoptotic proteins Bcl-2, Akt and mTOR in whole-tissue lysates of patients with BAV and TAV from the convex (V) and concave (K) regions of the aorta. (**A**) Relative Bcl-2 protein expression via Western blot analysis in the tissue of patients with BAV (*n* = 98) and TAV (*n* = 85) and separated by regions BAV V (*n* = 50), BAV K (*n* = 48), TAV V (*n* = 43) and TAV K (*n* = 42). α-tubulin served as the loading control. (**B**) Relative Akt protein expression via Western blot analysis in the tissue of patients with BAV (*n* = 103) and TAV (*n* = 84) and separated by regions BAV V (*n* = 53), BAV K (*n* = 50), TAV V (*n* = 43) and TAV K (*n* = 41). GAPDH served as the loading control. (**C**) Relative mTOR protein expression via Western blot analysis in the tissue of patients with BAV (*n* = 94) and TAV (*n* = 95) and separated by regions BAV V (*n* = 49), BAV K (*n* = 45), TAV V (*n* = 50) and TAV K (*n* = 45). GAPDH served as the loading control. *p* values of Student’s *t*-test: * *p* < 0.05, ** *p* < 0.01.

**Figure 4 ijms-24-07429-f004:**
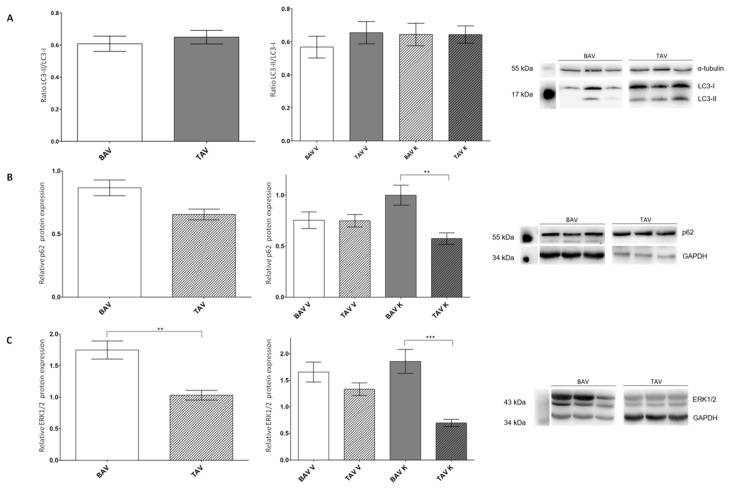
Statistical analysis of protein expression levels of LC3, p62 and ERK1/2 in whole-tissue lysates of patients with BAV and TAV from the convex (V) and concave (K) regions of the aorta. (**A**) Ratio of LC3II/LC3-I protein expression via Western blot analysis in the tissue of patients with BAV (*n* = 56) and TAV (*n* = 67) and separated by regions BAV V (*n* = 27), BAV K (*n* = 29), TAV V (*n* = 34) and TAV K (*n* = 32). α-tubulin served as the loading control. (**B**) Relative p62 protein expression via Western blot analysis in the tissue of patients with BAV (*n* = 193) and TAV (*n* = 94) and separated by regions BAV V (*n* = 49), BAV K (*n* = 44), TAV V (*n* = 49) and TAV K (*n* = 46). GAPDH served as the loading control. (**C**) Relative ERK1/2 protein expression via Western blot analysis in the tissue of patients with BAV (*n* = 93) and TAV (*n* = 99) and separated by regions BAV V (*n* = 50), BAV K (*n* = 43), TAV V (*n* = 52) and TAV K (*n* = 47). GAPDH served as the loading control. *p* values of Student’s *t*-test: ** *p* < 0.01, *** *p* < 0.0001.

**Figure 5 ijms-24-07429-f005:**
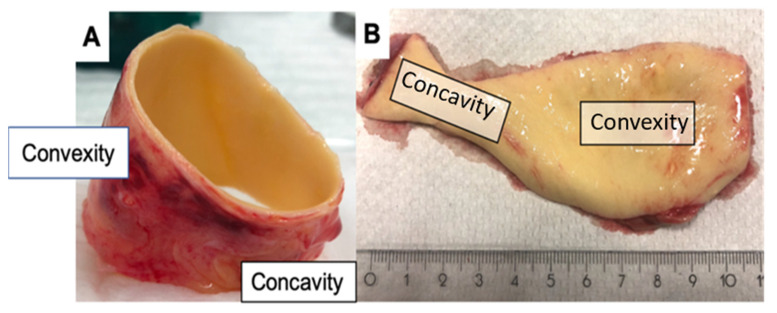
Excised aortic specimen. (**A**) Convexity (V) and concavity (K) region in total. (**B**) Convexity (V) and concavity (K) region after preparation.

**Table 1 ijms-24-07429-t001:** Results and amounts (*n*) of the relative protein expressions of RAGE and AGE generated using Western blot analysis and protein expression of S100A6 in pg/mL/100 µg generated using ELISA in whole-tissue lysates of patients with BAV and TAV from the convex (V) and concave (K) regions of the aorta. All results are shown as mean ± SEM.

	BAV V + K	TAV V + K	BAV V	TAV V	BAV K	TAV K
**RAGE** ** *n* **	1.59 ± 0.14 69	1.07 ± 0.08 59	1.26 ± 0.16 33	1.25 ± 0.10 30	1.89 ± 0.21 36	0.89 ± 0.13 29
**AGE** ** *n* **	1.46 ± 0.10 69	0.95 ± 0.09 59	1.34 ± 0.13 33	1.00 ± 0.11 30	1.53 ± 0.13 36	0.84 ± 0.12 29
**S100A6** ** *n* **	2154 ± 155 57	1610 ± 167 89	1889 ± 197 29	1633 ± 270 42	2429 ± 235 28	1589 ± 207 47

**Table 2 ijms-24-07429-t002:** Results and amounts (*n*) of the ratio of relative protein expressions of active caspase-3 (17 kDa; 12 kDa) and mature inactive caspase-3 (32 kDa), as well as relative protein expressions of vimentin and α-smooth muscle actin in whole-tissue lysates of patients with BAV and TAV from the convex (V) and concave (K) regions of the aorta generated via Western blot analysis. All results are shown ± SEM.

	BAV V + K	TAV V + K	BAV V	TAV V	BAV K	TAV K
**active/mature caspase-3** ** *n* **	1.21 ± 0.09 97	1.56 ± 0.12 87	1.36 ± 0.17 50	1.40 ± 0.14 42	1.25 ± 0.13 47	1.83 ± 0.22 45
**vimentin 57 kDa** ** *n* **	2.13 ± 0.26 99	2.18 ± 0.25 94	2.38 ± 0.46 49	2.13 ± 0.39 48	1.89 ± 0.25 50	2.23 ± 0.30 46
**vimentin 48 kDa** ** *n* **	3.20 ± 0.34 96	2.73 ± 0.27 93	3.21 ± 0.49 48	2.33 ± 0.32 46	3.19 ± 0.49 48	3.11 ± 0.44 47
**α-smooth muscle actin** ** *n* **	1.51 ± 0.10 104	1.07 ± 0.09 86	1.44 ± 0.14 52	1.10 ± 0.11 44	1.59 ± 0.14 52	1.04 ± 0.14 42

**Table 3 ijms-24-07429-t003:** Results and amounts (*n*) of the relative protein expressions of the anti-apoptotic proteins Bcl-2, Akt and mTOR in whole-tissue lysates of patients with BAV and TAV from the convex (V) and concave (K) regions of the aorta generated via Western blot analysis. All results are shown ± SEM.

	BAV V + K	TAV V + K	BAV V	TAV V	BAV K	TAV K
**Bcl-2** ** *n* **	1.22 ± 0.07 98	1.38 ± 0.08 85	1.36 ± 0.17 50	1.33 ± 0.1 43	1.10 ± 0.10 48	1.42 ± 0.12 42
**Akt** ** *n* **	1.28 ± 0.08 103	1.29 ± 0.09 84	1.26 ± 0.10 53	1.20 ± 0.11 43	1.25 ± 0.11 50	1.40 ± 0.14 41
**mTOR** ** *n* **	0.81 ± 0.06 94	0.64 ± 0.05 95	0.74 ± 0.08 49	0.71 ± 0.07 50	0.88 ± 0.08 45	0.58 ± 0.07 45

**Table 4 ijms-24-07429-t004:** Results and amounts (*n*) of the ratio of the relative protein expressions of LC3-II and LC3-I and the relative protein expression levels of p62 and ERK1/2 in whole-tissue lysates of patients with BAV and TAV from the convex (V) and concave (K) regions of the aorta generated via Western blot analysis. All results are shown ± SEM.

	BAV V + K	TAV V + K	BAV V	TAV V	BAV K	TAV K
**LC3-II/LC3-I** ** *n* **	0.61 ± 0.05 56	0.65 ± 0.04 67	0.57 ± 0.07 27	0.64 ± 0.07 34	0.65 ± 0.07 29	0.64 ± 0.05 32
**p62** ** *n* **	0.87 ± 0.06 93	0.66 ± 0.04 94	0.75 ± 0.08 49	0.75 ± 0.06 49	1.00 ± 0.10 44	0.58 ± 0.06 46
**ERK1/2** ** *n* **	1.75 ± 0.14 93	1.03 ± 0.08 99	1.66 ± 0.19 50	1.33 ± 0.12 52	1.86 ± 0.23 43	0.70 ± 0.07 47

**Table 5 ijms-24-07429-t005:** Patients’ characteristics. Listed are the total amounts of patients with BAV and TAV, average age in years ± SEM, sex, diameter of the aorta at the time of surgery in mm ± SEM and total amount and percentage of patients suffering from aortic regurgitation, aortic stenosis, diabetes, coronary heart diseases and arterial hypertension.

	BAV (*n* = 57)	TAV (*n* = 49)
**age (year)**	61.8 ± 11.8	65.5 ± 11.6
**male**	50 (88%)	36 (73%)
**diameter (mm)**	49.4 ± 3.9	51.1 ± 10.6
**aortic regurgitation**	19 (33%)	40 (82%)
**aortic stenosis**	33 (58%)	5 (10%)
**diabetes**	3 (5%)	3 (6%)
**coronary heart diseases**	6 (11%)	9 (18%)
**arterial hypertension**	48 (84%)	43 (88%)

## Data Availability

All data presented are available upon request.

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
