# Peer review of "Apoptotic Cell Death in Bicuspid-Aortic-Valve-Associated Aortopathy"

_ijms, 2023, doi:10.3390/ijms24087429_

Round 1
Reviewer 1 Report
The manuscript is potentially interesting and well designed. The Authors investigated the role of apoptosis and autophagy in bicuspid aortic valve associated aortopathy. Experiments are well described as well as data presented. Only few correction should be added into revised version of manuscript.
Comments:
- Please, clarify in Materials and Methods section that how samples were collected (‘n’ numbers). As demonstrated, Table 1 “Listed are the total amounts of patient with BAV and TAV,..” there are some paradox information in case of Fig 2, 3,4, 5 about ‘n’ numbers. Please, make it clear.
- Section ‘Protein expression of pro-apoptotic proteins’: Figure 3A shows the ratio of the active caspase-3 and mature caspase-3. What does it mean precisely? Please, define it.
- The Authors applied GAPDH as housekeeping protein for moralization. Please, mention it in Materials and Methods section that how the results were calculated. Moreover, on the most of figures the intensity of GAPDH seems to fluctuating. There is evident differences between BAV and TAV samples. Could you explain it?
- Section: Investigation of autophagy-linked LC3 and p62:
This guideline (Guidelines for the use and interpretation of assays for monitoring autophagy (3rd edition). Autophagy. 2016;12(1):1-222. doi: 10.1080/15548627.2015.1100356.) suggests that, there is not always a clear precursor/product relationship between LC3-I and LC3-II. Have the Authors evaluated only the level of LC3-II in the tissue of patients with BAV and TAV V/K?
Reviewer 2 Report
The strength of this paper is that it contains data on a large series of patients with aortic aneurysm in the presence of a bicuspid aortic valve or a tricuspid aortic valve. The authors have diligently executed analyses on aortic tissue. My main concern is the lack of a coherent message.
- Please reduce the number of non-standard abbreviations.
- The authors analyzed ascending aortic tissues of patients with aortic aneurysm and bicuspid aortic valve or tricuspid aortic valve morphology with regard to selected proteins of signal transductions pathways. These are all patients with aortic aneurysm for the obvious reason that aortic tissue is required. This is a highly selected population of subjects that all have an aortopathy. What was then the larger objective of the study that goes beyond molecular and cellular details?
- Autophagy was originally characterized as a cell survival mechanism for amino acid recycling during starvation. Whether autophagy functions primarily in cell survival or cell death is a critical question yet to be answered. Is it not more appropriate to use the term ‘autophagic cell death or type 2 cell death’ in the context of this paper?
- The abstract should be clearer in terms of providing quantitative data that illustrate the main findings of the study. It should contain more specific and less descriptive information. The reader really wants to know what are the salient findings of the study and what is the message. There is not sufficient coherence in the abstract.
- After reading the paper it is still not clear to me what is the mode of cell death underlying ‘increased cell death in aortic tissue of BAV patients’. It is apparently certainly not apoptosis and not autophagic cell death. Once again, autophagy can also be a mechanism of cell survival.
Round 2
Reviewer 1 Report
Accepted
Author Response
We thank the reviewer for his/her time and effort.
Reviewer 2 Report
After reading the response of the authors, I am still lacking a coherent message.
1. The authors demonstrate increased cell death in aortic tissue of BAV patients compared to TAV patients. This appears to be the main observation. How do you define cell death considering that there are three modes of cell death?
2. After reading the clarifications of the authors, I come to the conclusion that the title of the paper tends to be misleading. There is too much uncertainty.
3. The strength of the study is that the authors have a large set of tissues obtained from aneurysm patients with either BAV or TAV. Even when the authors present 'preliminary' data, it is still required that part of the paper concerns clear and meaningful data that can be interpreted.
Round 3
Reviewer 2 Report
The authors have provided a satisfactory response to my comments.